evolution, theoretical biology, health and disease and epidemiology

trade-offs, host–parasite interactions, coevolution, virus evolution, niche breadth, theory

**Author for correspondence:**
Elisa Visher
e-mail: elisa_visher@berkeley.edu

# The problem of mediocre generalists: population genetics and eco-evolutionary perspectives on host breadth evolution in pathogens

Elisa Visher[1] and Mike Boots[1,2]

[1]Department of Integrative Biology, University of California Berkeley, Berkeley, CA, USA
[2]College of Life and Environmental Sciences, University of Exeter, Cornwall Campus, Ringgold Standard Institution, Penryn, Cornwall, UK

EV, 0000-0003-3984-4748; MB, 0000-0003-3763-6136

Many of our theories for the generation and maintenance of diversity in nature depend on the existence of specialist biotic interactions which, in host–pathogen systems, also shape cross-species disease emergence. As such, niche breadth evolution, especially in host–parasite systems, remains a central focus in ecology and evolution. The predominant explanation for the existence of specialization in the literature is that niche breadth is constrained by trade-offs, such that a generalist is less fit on any particular environment than a given specialist. This trade-off theory has been used to predict niche breadth (co)evolution in both population genetics and eco-evolutionary models, with the different modelling methods providing separate, complementary insights. However, trade-offs may be far from universal, so population genetics theory has also proposed alternate mechanisms for costly generalism, including mutation accumulation. However, these mechanisms have yet to be integrated into eco-evolutionary models in order to understand how the mechanism of costly generalism alters the biological and ecological circumstances predicted to maintain specialism. In this review, we outline how population genetics and eco-evolutionary models based on trade-offs have provided insights for parasite niche breadth evolution and argue that the population genetics-derived mutation accumulation theory needs to be better integrated into eco-evolutionary theory.

## 1. Introduction

Why specialists exist in the face of broad-niched generalists remains a central question for both ecology and evolution. At a fundamental level, this question underpins most of our theories for why there is so much genetic, phenotypic and species diversity in nature [1,2]. The predominant explanation in the literature is that this coexistence is maintained by costs to generalism such that specialists can coexist with or outcompete generalists in at least some contexts [3]. The concept of costly generalism has been shown to be important for explaining many ecological and evolutionary processes, including abiotic adaptive radiations [4,5], diversity in tropical forests [6,7], co-diversification [8] and macroevolutionary patterns of diversity [9]. There has also been considerable interest in the importance of *niche breadth* evolution in host–pathogen interactions as these systems may be important drivers of diversity [10,11] and are relevant for multi-host disease transmission and zoonosis [12].

For these reasons, the study of pathogen *niche breadth* (box 1) evolution has flourished in several different subfields including evolutionary genetics, epidemiology and ecology [12–15]. In this review, we focus on integrating insights for parasite *niche breadth* evolution from theoretical and empirical perspectives inspired by both population genetics and eco-evolutionary theory, focusing particularly on virus

**Box 1.** Glossary of terms.

| | |
|---|---|
| parasite and pathogen | parasite and pathogen both refer to organisms that are dependent on hosts for replication and cause host damage; we have used these terms interchangeably. We use virus when talking about empirical virus data or concepts that depend on nuances of viral biology |
| niche breadth | the range of environments an organism is adapted to |
| antagonistic pleiotropy | when one allele has positive effects for one fitness component and negative for another |
| spatial heterogeneity | environmental patches differ |
| temporal heterogeneity | an individual environmental patch changes over time |
| coarse-grained environment | an individual experiences a constant environment, but its offspring experience a different constant environment |
| fine-grained environment | an individual organism will experience all values of heterogeneity in its lifetime |
| pareto front | the optimal trade-off front dividing accessible, suboptimal phenotype space from phenotype space that is inaccessible owing to constraints |
| susceptible, infected, recovered (SIR) model | a type of compartmental model that tracks susceptible, infected and recovered individuals and the movement between such classes |
| evolutionary stable strategy (ESS) | when the eco-evolutionary models has a singular strategy at ecological equilibrium that is uninvadable |
| branching | when the eco-evolutionary model has multiple stable strategies, such that the population is predicted to split into two or more phenotypes |
| cycling | when the model has a shifting stable strategy such that the populations' phenotypes cycle over time |
| fitness landscape | the fitness of possible genotypes which can include global and local fitness peaks |
| genetic drift | the impact of stochastic processes on mutational frequencies |
| genetic hitchhiking | when neutral or deleterious sequence variants increase in frequency because an allele linked to them is selected for |
| clonal interference | when beneficial mutations arise on different genotypes such that their populations are in competition with each other |
| epistasis | when one gene affects the expression of others |

evolution. These subfields' broad perspectives towards infectious disease evolution differ in several key ways and are not often well integrated, especially across intra- and inter-host scales [16,17]. Both population genetics and eco-evolutionary theory have important insights for why generalists are not ubiquitous, but these perspectives have not been well combined for a unified understanding of viral *niche breadth* evolution [18–23], though see [24].

Both bodies of theory have considered how *niche breadth* evolves when there are costs to expanded host range owing to direct trade-offs or *antagonistic pleiotropy* [19,25,26]. However, population genetics theory has expanded to explore other mechanisms of costly generalism, such as mutation accumulation [23]. As such, there exists a large gap in the literature on how non-trade-off mechanisms of costly generalism might function in eco-evolutionary contexts. Therefore, we aim to summarize some of the fundamental assumptions of viral eco-evolutionary and population genetics theory, outline their insights on *niche breadth* evolution and highlight gaps where these perspectives' different insights should be united to understand *niche breadth* dynamics in broader ecological and evolutionary contexts.

## 2. Why is not everything a generalist?

Heterogenous environments are a universal feature of any natural ecosystem and any number of abiotic or biotic environmental dimensions can be heterogenous. Some of the earliest theoretical considerations for how environmental heterogeneity affects evolution come from Levins [19]. The fundamental question of this early work was: why is not every species an environmentally flexible generalist in the face of ubiquitous environmental heterogeneity? Essentially, if generalists can use a wider array of resources, then why do they not outcompete specialists? The continuing presence of specialists means that there must be some cost of adapting to different environments [19].

Levins [19] used models to examine how costs to generalism could select for specialists in different conditions of environmental heterogeneity. In these models, Levins imposed a trade-off between fitness on different environments. Though he was agnostic to the mechanism of such a trade-off, his model generally considers direct phenotypic trade-offs caused by *antagonistic pleiotropy*. The models considered different types of environmental heterogeneity including *spatial* or *temporal* and 'coarse' grained or 'fine' grained. The type of heterogeneity, environmental 'grain' and shape of the trade-off determine whether an organism is selected to generalize or to specialize on a subset of the environment. With *temporal* environmental heterogeneity, costs to generalism can select for single (*fine grained*) or multiple (*coarse grained*) specialists [19].

## 3. Do trade-offs actually exist?

Trade-off theory dominated many explanations for *niche breadth* evolution, but this theory began to be questioned as groups started explicitly trying to measure trade-offs in traits

underlying adaptation to different environments (reviewed in [27,28]). They found that trade-offs are sometimes identifiable in experimental and natural systems, but are far from universally observed [27,28]. Indeed, when host breadth has been measured in viruses, a wide body of empirical literature finds mixed results for whether viral fitness tradeoff across host species and host genotypes (reviewed in [22,29]).

Some of this failure to find trade-offs may be because they are difficult to measure [30]. To test for trade-offs, populations are often measured in simple, high-resource laboratory environments where fitness costs may be obscured [31]. If trade-offs are multi-dimensional, fitness costs must also be tested for in many traits [30,32]. Even when no-cost generalism is observed, this could be a transient state owing to populations not yet being adapted to laboratory conditions. Li *et al.* [33] showed that metabolic functions in yeast trade off with specialists having the highest performance on each metabolic function and an optimal trade-off front (i.e. *Pareto front*) defining a region of inaccessible no-cost generalist parameter space. They also showed that many genotypes can exist below the *Pareto front*. In this case, no-cost generalist alleles could advance the fitness of these maladapted genotypes to *Pareto front*, but evolution would be determined by the trade-off afterwards [33,34]. Accordingly, Satterwhite & Cooper [35] show that no-cost generalism may be temporary with generalists able to adapt to multiple resources as fast as specialists at the beginning of evolution, but lagging over larger time scales.

Regardless of these difficulties in measuring trade-offs, measurements of the correlation between viral fitness on different hosts have ranged from negative, to neutral, to positive [22,29]. Therefore, trade-offs seem to function in at least some systems, but the persistent challenge of measuring them probably means that there are additional mechanisms maintaining specialists [22]. Empirical results do suggest that experimental evolution along genetically diverse hosts can constrain the evolution of virulence or the rate of adaptation [36–40].

## 4. Introduction to modelling frameworks

### (a) The population genetics perspective

Population genetics models of pathogen evolution focus on how the availability of mutations and the ability of selection to act on them together determine fitness outcomes [41]. Fitness in these models is generally considered to be a static *fitness landscape* that can be a simple peak or more rugged depending on the genetic architecture of the system and the number of possible strategies to respond to an environmental pressure [42,43]. Population genetics models of pathogen evolution tend to follow pathogen replication directly, so that parasite fitness is maximized at the highest replication rate (figure 1). Viral evolutionary models following this framework are especially concerned with the high mutational rates and population sizes of RNA viruses in particular [44].

These models consider mutational fitness effects and the selection pressures on them that, depending on their strength relative to *genetic drift*, can lead to the purging, balancing or fixation of such new mutations [53]. They also consider how variation in time to fixation (or purging) can affect processes like *genetic hitchhiking* and *clonal interference* [54,55] and how genotypes can vary in evolvability to shifting environments [56,57]. Through these processes, population genetics theory explores

the ability of a genome to reach any predicted fitness 'peak' and its ability to preserve any advantageous phenotype.

### (b) The eco-evolutionary perspective

Eco-evolutionary models incorporate explicit ecological feedbacks into evolutionary models to consider how invading phenotypes reshape the ecological equilibrium of a system and therefore potentially change the optimal fitness strategy [58,59]. For hosts and parasites, many eco-evolutionary models are built around epidemiological *SIR* type compartmental models which follow the infection, recovery, birth and death of host individuals [60]. This means that these models track transmission between these compartments, rather than pathogen replication directly (figure 1) [61].

Generally, these models consider quantitative trait phenotypes (or strategies) and their trade-offs rather than explicitly including complex genetic or mutational processes, and follow frameworks where the invasion of any new strategy affects the equilibrium population sizes of each organism in the system [58,62,63]. Therefore, these models allow for the exploration of dynamic fitness peaks that shift with ecological conditions and may lead to *evolutionarily* and *coevolutionarily stable strategies, branching* or *cycling* [64,65].

## 5. How does *niche breadth* evolve when trade-offs drive costly generalism?

Modern trade-off theory assumes that costs to generalism are owing to *antagonistic pleiotropy*, where mutations conferring fitness on one environment hinder fitness on others [66]. In both population genetics and eco-evolutionary approaches, models based on trade-offs have been extended across different ecological and coevolutionary conditions to make predictions about the effects of host ecology and genetics on *niche breadth* and diversity dynamics.

### (a) Trade-offs in population genetics theory

In population genetics theory, the main model for host range in host–pathogen systems is the gene-for-gene (GfG) model, where *niche breadth* is determined by the collection of virulence and resistance alleles [67]. In this model, infectivity and resistance ranges directly trade off with pathogen replication and host reproduction, respectively [68,69]. These GfG models predict that parasites and hosts of different range breadths will either stably coexist or cycle depending on the number of loci in the system and their costs [26,69].

However, classic GfG models only allow fluctuations in range breadth (i.e. the number of hosts that a parasite infects or the number of parasites that the host resists) [70]. This is because they imply that the resistance and infectivity ranges of their hosts and parasites are entirely nested so that the least infective parasite only infects the most permissive host and the most resistant host can only be infected by the most infective, generalist parasite. Instead, GfG interactions may coexist with matching allele interactions with multiple specialists on a spectrum or as part of a two-part process to allow for rare genotype advantage [71,72]. When the assumption of complete nestedness is relaxed so that there can be multiple identically ranged hosts and parasites infecting different subsets of the total population, both high-frequency cycling between hosts and parasites

**Figure 1.** Integrating theory across scales of viral fitness. 1, [44]; 2, [45]; 3, [46]; 4, [47]; 5, [48]; 6, [21]; 7, [49]; 8, [50]; 9, [51]; 10, [10]; 11, [52]. Created with Biorender.com. (Online version in colour.)

matching different subsets of the range and lower frequency cycling in range breadth can occur [70]. In other words, identities of the hosts and their matching parasites in the system will cycle quickly owing to rare genotype advantage and the average range breadth of the hosts, and their parasites will cycle more slowly owing to costs of range breath. However, population genetics models typically assume constant large population sizes and extending these GfG models to include ecology shows that demographic and ecological factors can strongly affect dynamics of a system [70,73–75]. Allowing fluctuations in population size disrupts the regularity of cycling and can lead to dampening over longer time scales.

## (b) Trade-offs in eco-evolutionary theory

In the eco-evolutionary theory on infectious diseases, trade-offs have been extensively applied by combining evolutionary trade-off models and epidemiological (ecological) models [13,20,21,76]. These models have shown that specialists will evolve when the shape of the trade-off between two host types is negative and accelerating (i.e. the generalist is less than half as fit on either host as the specialist), regardless of whether this trade-off is on virulence or transmission [21,76]. These models generally assume symmetry so that there is either one specialist per host or a single generalist population, but empirical results have also shown that the presence of a host without a specialist (i.e. an empty niche) can select for

**Box 2.** Empirical tests of the relative importance of trade-off and mutation accumulation theories.

A few experiments have directly considered whether mutation accumulation or trade-off mechanisms for costly generalism are more important in their systems. Cooper & Lenski [80] test this question in the long-term evolution experiment (LTEE) after bacteria populations were selected on novel resource for 20 000 generations. They show that the evolution of novel resource use was correlated with a high loss of function on other environments soon after the gain of novel function, as would be predicted by *antagonistic pleiotropy*. They also consider the experiment's high mutation rate lineages to show that these lines did not have larger fitness losses on alternate environments, as would have been predicted by mutation accumulation theories [80]. However, after 50 000 generations of evolution, Leiby & Marx [81] re-assayed these LTEE lines using a different growth assay across a broader range of nutrients. They found that high mutation rate lineages suffered higher fitness decreases on alternate environments only after 50 000 generations, suggesting that the differences between the lineages were not yet apparent at the earlier time point and that mutation accumulation drives resource specialization in the LTEE.

In a separate experiment, Remold *et al.* [82] showed that the relative importance of *antagonistic pleiotropy* and mutation accumulation may depend on the specific environments that the population is adapting to, as well as the population's evolutionary history. They used previously evolved virus lines from Turner & Elena [15] where viral populations were evolved on only MDCK or only HeLa host cells or alternated between the two cell types. Remold *et al.* showed that the trade-off between host types seen in single host evolved lines was best explained by *antagonistic pleiotropy* when they evolved on one of the cell lines and by mutation accumulation when they evolved on the other cell line. Additionally, Turner & Elena [15] and Remold *et al.* [82] showed that viruses evolved on alternating host lines were able to increase their fitness on both host types, indicating that alternating host treatments were able to select for 'costless' mutations, conferring advantages on both host types.

In a meta-analysis, Bono *et al.* [29] concluded that the evolution of strategies with trade-offs rather than costless generalism can be partially predicted by the selective environment of experimental evolution. Trade-offs are most often found when populations have been evolved in homogeneous environments (versus heterogenous), in *spatially heterogenous* environments (versus *temporally heterogenous*), or in longer experiments (versus ones that have been evolved for fewer generations). These factors all suggest that trade-offs exist, but that they are most consistently evolved when populations adapt to selective environments that allow them to experience one environment. Meanwhile, populations experiencing selective environments where they are forced to replicate on both environments often find mechanisms of adaptation that do not include trade-offs. The trend of trade-offs being found more often in longer experiments suggests that mutation accumulation might be a common driver of costs to generalism, that short-term evolution may be dominated by mutations below *Pareto fronts* or that selection from standing variation is likely to select for less pleiotropic alleles than selection from novel mutation [83].

---

generalists [77,78]. However, this benefit is likely to be temporary as empty niches are likely to be filled by their own specialists [78].

Other models have also used mixing matrices to show that higher between host type (versus within-host type) transmission selects more for generalists, even when generalism is costly [13,20]. Taken together, these theoretical studies convincingly show that the evolution of parasite *niche breadth* in heterogeneous host environments can be determined by a combination of the shape of the trade-off and the contact structure between the between host types.

It is also clear that hosts commonly evolve pathogen resistance and that different ecological dynamics between host and pathogen populations are likely to create varying selection pressures for resistance evolution and, therefore, coevolutionary feedbacks [79]. Best *et al.* [25] show that a coevolutionary range model, where resistance and infectivity breadths are nested and costly, will generate and maintain stable diversity with coexisting hosts and parasites across the generalism–specialism range. Essentially, this occurs because resistance–infectivity range matching is a *stable strategy* for the parasites, but an unstable one for the hosts. This leads to hosts *branching* into higher resistance (lower infection) and lower resistance (higher reproduction) strains, which the parasites subsequently partition. In other words, a host whose resistance breadth exactly overlaps with a parasite's infectivity breadth should either evolve further costly resistance to prevent infection from that parasite or evolve less resistance to reproduce faster, as it is susceptible anyway. Furthermore, Boots *et al.* [10] show how changing the trade-off

shape in this model can lead the system to maintain dimorphic strains, multiply branch and maintain stable diversity, or cycle between range breadths. A key driver of the maintenance of stable diversity with coexisting hosts and parasites across the generalism–specialism range is the existence of incompatibilities between host–parasite pairs.

## 6. How does *niche breadth* evolve when non-trade-off mechanisms drive costly generalism?

Because it has been difficult to measure trade-offs between hosts and environments, parts of modern population genetics theory have focused on proposing costs to generalism beyond strict genetic *antagonistic pleiotropy* (box 2). First, Fry [84] showed that specialism could be maintained if alleles that are strongly beneficial in one environment are neutral or weakly beneficial in others and the host could choose its environment. Effectively, if there is a benefit to specialization compared to the ancestor, there does not need to be a cost to generalism.

More generally, population genetics theory has explained the existence of specialists and generalists through mutational and selection processes. Kawecki [18] and Whitlock [23] argue that generalists could have lower fitness because organisms in *coarse-grained* environments experience selective pressure from only one environment a generation. Because of this, generalists have less selective pressure to fix beneficial mutations or purge deleterious mutations that are specific to any one environment. Under this theory, a generalist could

**Box 3.** Current challenges in integrating eco-evolutionary and population genetics models across scales.

Recently, much attention has been turned to integrating theory across scales of biological organization [51]. Both eco-evolutionary and population genetics frameworks for pathogen evolution have major limitations. Most simply, eco-evolutionary perspectives are poorly equipped to consider the selective ability to reach fitness peaks, while population genetics perspectives are poorly equipped to explore ecological feedbacks and dynamic fitness peaks.

Eco-evolutionary models have not yet well explained how selective strategies maintained by population-level ecological feedbacks are actually maintained in the face of high mutation rates and short-term evolution [93,94]. In these models, the trade-offs that shape evolutionary strategies sometimes occur across very different temporal scales, perhaps leading to differing mutation and selection pressures on the different traits in the trade-off.

On the other hand, many theoretical and experimental treatments of population genetic processes in viruses consider within-host processes or selection through cell culture [16]. However, recent work in natural host–pathogen systems has shown increasing consideration about how these within-host evolutionary processes translate to selection between hosts through the extremely tight, *drift* promoting bottlenecks of many natural transmission events [49,95–97]. Therefore, it is unclear how many of the processes deemed important in population genetics models function through transmission events and affect population-level evolution and lineage-level selection.

Therefore, better integrating these two fields may help us better understand how population genetics strategies function at the level of between host transmission and lineage-level selection and how eco-evolutionary feedbacks are selected through shorter term mutational and selection processes.

technically be just as fit as a specialist on each environment, but its fitness quickly decays because it cannot continuously be selected for fitness on all environments.

However, one important caveat is that this specialist advantage in fixation probability can be offset if generalists have population sizes that increase linearly in relation to their *niche breadth* because the availability of new mutations is modulated by population size [23]. While the fixation of beneficial alleles is then equal in small specialist and large generalist populations, the speed of fixing these beneficial alleles is much slower in large generalist populations. Therefore, these generalists may be less able to track changing environments and may suffer from costs associated with *genetic hitchhiking* and *clonal interference* [23,85]. This process can be sped up by limited dispersal and the ability for organisms to choose their environment [86].

Mutation accumulation theories explain how generalism can be costly even without genetic trade-offs. However, no-cost generalism has been shown in many experiments so additional theory is needed to explain why specialism occurs in the face of seemingly no-cost generalism. Recently, there has been increasing discussion about how viral *niche breadth* affects future evolvability because of the differing genetic constraints on specialists and generalists [52]. Remold [22] proposes that *epistatic* pleiotropy may impede the evolvability of viruses. Theory and experimental work suggests that because there are more genetic options for producing specialists than creating no-cost generalists, specialist populations will maintain higher genetic diversity with less genetic constraints to evolve to new selection pressures [52,83,87]. However, macroevolutionary trends suggest that specialist lineages have higher extinction rates than generalists and prefer to specialize on more stable environments [88–90].

# 7. Ecological implications of multiple mechanisms of costly generalism

While trade-off and mutation accumulation theories for the evolution of specialization have often been presented as in conflict with each other, they both start with the observable phenomenon that specialists exist and that generalists are often less adapted to any one environment. Fundamentally, trade-off models start with the observation that there is a cost to generalism and explain how ecological processes may select for different *niche breadth*s, while mutation accumulation models propose additional explanations for mediocre generalism. In a 2005 review, Maclean [4] briefly considers trade-off and mutation accumulation models and suggests that there may not actually be a conflict between these theories if we consider that trade-offs may occur not only as a result of negative genetic correlations, but also evolve as an indirect consequence of mutation accumulation during specialization. In other words, the accumulation of environmentally specific deleterious alleles could result in fitness correlations that function analogously to genetic trade-offs in eco-evolutionary contexts. This begs the question of whether such a trade-off created by selection pressures would itself shape further ecology and evolution in a way that qualitatively differs from trade-offs owing to *antagonistic pleiotropy*. Theory needs to be developed to determine whether models based upon these two mechanisms (trade-offs and mutation accumulation) differ in ways that would affect how different population structures select for *niche breadth* evolution, how coevolution shapes these traits, and how *niche breadth*s selected by these two mechanisms may impact population-level dynamics and diversification processes. Thus, these models' separate insights could be combined to generate integrated theory of *niche breadth* evolution.

Comparing these classes of models is obviously complicated by the different underlying assumptions of modelling methods. A combined theoretical framework would have to allow a trade-off that is shaped by population genetics models of mutation and selection to feed into an eco-evolutionary model similar to those based on trade-offs to predict selection for *niche breadth*. In this combined model, any trade-off curves may emerge through the balance between mutational and purging pressures rather than underlying assumptions of *antagonistic pleiotropy*. Such models would need to rely heavily on simulation but should be tractable and allow clear insights as population genetic and eco-evolutionary theoretical frameworks have already been developed. More generally, similar approaches should be developed to better integrate population

genetics and eco-evolutionary theory insights across evolutionary biology.

## 8. Conclusion

*Niche breadth* evolution in parasites is an important problem that a large body of empirical work and theory has attempted to explain. Both population genetics and eco-evolutionary theory have important insights on the topic, but these have not yet been well integrated to produce a unifying theory of how both genetic and ecological processes might interact with pathogen biology to shape *niche breadth* evolution. We argue that further theoretical work is needed to unify these perspectives to predict how *niche breadth* evolves and how various ecological conditions may bias selection towards generalism or specialism. Unifying these perspectives to explore how non-trade-off costs of generalism shape the coevolution of hosts and pathogens in eco-evolutionary models would allow us to determine whether the mechanism of costly generalism shapes *niche breadth* coevolutionary dynamics in ways that alter our understandings about the ecological conditions predicted to drive spillover and maintain diversity. It is likely that differences, if they arise, are likely to depend on temporal scale and open up further avenues of research regarding the role of temporal scale and habitat instability in *niche breadth* diversification dynamics [91,92].

Finally, the lack of unification across population genetics and eco-evolutionary perspectives is not limited to *niche breadth* evolution. A wider consideration of the exact nature of trade-offs could help understand how selective pressures are maintained across scales (box 3). For example, the addition of population genetics processes into evolutionary epidemiology models on the virulence and transmission trade-off provides predictive insights beyond those generated by simple trade-off functions [98,99]. This may be particularly important because the pathogen particles that transmit are not necessarily the same ones causing host damage [100], meaning that both the mean and the variance of the genotypes within the host determine the outcome of pathogenesis and transmission for the infection. In general, a better consideration of the nature of trade-offs in the light of constant mutation pressure and ecological feedbacks should help us better understand ecological and evolutionary processes.

Data accessibility. This article has no additional data.
Authors' contributions. E.V. and M.B. conceptualized and wrote the paper.
Competing interests. We declare we have no competing interests.
Funding. E.V. acknowledges support from an NSF GRFP DGE 1752814, and M.B. acknowledges support from NIH/R01-GM122061-03 and NERC NE/K014617/1.
Acknowledgements. We would like to thank members of the Boots Laboratory, Britt Koskella and Lisa Bono for helpful discussions.

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
