## [Reviewer comments · Proceedings of the Royal Society B: Biological Sciences]

Review History

RSPB-2020-1230.R0 (Original submission)

Review form: Reviewer 1

Recommendation

Accept with minor revision (please list in comments)

Scientific importance: Is the manuscript an original and important contribution to its field?

Excellent

General interest: Is the paper of sufficient general interest?

Good

Quality of the paper: Is the overall quality of the paper suitable?

Excellent

Is the length of the paper justified?

Yes

Should the paper be seen by a specialist statistical reviewer?

No

Do you have any concerns about statistical analyses in this paper? If so, please specify them explicitly in your report.

No

It is a condition of publication that authors make their supporting data, code and materials available - either as supplementary material or hosted in an external repository. Please rate, if applicable, the supporting data on the following criteria.

Is it accessible?

N/A

Is it clear?

N/A

Is it adequate?

N/A

Do you have any ethical concerns with this paper?

No

Comments to the Author

This manuscript reviews (mainly) the theoretical aspects of niche width evolution in host parasite systems. It focusses on viral infections, which in current times is highly relevant due to its direct implications in zoonosis. It's main strength is that it nicely illustrates the differences between eco-evolutionary and population genetic models, and calls for uniting eco-evolutionary models with population genetic theory. In general, I found reading this piece quite enjoyable and gained some new perspective on niche breadth evolution. Some terminology used and background knowledge expected is however rather addressed at the specialist and can probably leave the generalist a bit puzzled. I have made a few suggestions where this could be improved and have some other comments that can easily be included in a revision of the manuscript.

- 1.) Since I am not a theoretician, I cannot really judge the completeness of their model and theory overview. But I have the feeling that some ecological/demographic aspects of the genetic models for coevolution could be added. Gokhale, C.S., A. Papkou, A. Traulsen, and H. Schulenburg, Lotka-Volterra dynamics kills the Red Queen: population size fluctuations and associated stochasticity dramatically change host-parasite coevolution. *BMC Evolutionary Biology*, 2013. 13: 254. would make a valuable addition to the citation in l. 174.
- 2.) To me it seemed as if one assumption of most models is that they are somehow symmetric, i.e. there are 2 specialists (one for each host species) and 1 generalist. What happens when this symmetry is broken, i.e. the generalist can infect both host species, but there is only one specialist infecting one of the hosts? I could see that this would reduce the cost of generalism since many more infectable hosts are available.
- 3.) The conclusion seems to be rather short and reads more like a summary. Maybe adding a bit of outlook to what could be achieved practically from combining the different theories might give the whole article more weight?

Other suggestions:

-1195-201 While can guess the reasoning what is meant by 'resistance-infectivity range matching' and why this could be unstable for hosts, one or two explanatory sentences would really help to understand this important point.

-1199-201 it is a bit unclear what diversity is maintained here. I assume it is diversity of range breadth, but it could also be host and parasite diversity itself, which could also maintain range

breadth. Can the nestedness of both levels of diversity mad a bit clearer?

-1209 I wonder if a benefit to specialization is not the other side of the same cost as the cost to generalism, since fitness is measured realtive to each other. Can either exist without the other?

-box1: Could the 'where constraints between phenotypes emerge'-part of Pareto front be explained better? I have a hard time imagining what constraints between phenotypes are.

-1271 I somehow miss a section that details the rich body of empirical work on niche breadth. Box 2 only gives examples on trade-off and mutation accumulation, while the empirical data largely comes from other reviews. Maybe giving some more examples of how cost of generalism varies in detail for some systems might be helpful.

Review form: Reviewer 2

Recommendation

Accept with minor revision (please list in comments)

Scientific importance: Is the manuscript an original and important contribution to its field?

Excellent

General interest: Is the paper of sufficient general interest?

Excellent

Quality of the paper: Is the overall quality of the paper suitable?

Excellent

Is the length of the paper justified?

Yes

Should the paper be seen by a specialist statistical reviewer?

No

Do you have any concerns about statistical analyses in this paper? If so, please specify them explicitly in your report.

No

It is a condition of publication that authors make their supporting data, code and materials available - either as supplementary material or hosted in an external repository. Please rate, if applicable, the supporting data on the following criteria.

Is it accessible?

N/A

Is it clear?

N/A

Is it adequate?

N/A

Do you have any ethical concerns with this paper?

No

Comments to the Author

Please see the attached file. (See Appendix A)

Decision letter (RSPB-2020-1230.R0)

14-Jul-2020

Dear Dr Visher

I am pleased to inform you that your manuscript RSPB-2020-1230 entitled "The Problem of Mediocre Generalists: Population Genetics and Eco-Evolutionary Perspectives on Host Breadth Evolution in Pathogens" has been accepted for publication in Proceedings B.

The referees have recommended publication, but also suggest some minor revisions to your manuscript. Therefore, I invite you to respond to their comments and revise your manuscript. Because the schedule for publication is very tight, it is a condition of publication that you submit the revised version of your manuscript within 7 days. If you do not think you will be able to meet this date please let us know.

Online supplementary material will also carry the title and description provided during submission, so please ensure these are accurate and informative. Note that the Royal Society will not edit or typeset supplementary material and it will be hosted as provided. Please ensure that

the supplementary material includes the paper details (authors, title, journal name, article DOI). Your article DOI will be 10.1098/rspb.[paper ID in form xxxx.xxxx e.g. 10.1098/rspb.2016.0049].

Best wishes,
Innes Cuthill

Prof. Innes Cuthill
Reviews Editor, Proceedings B
mailto: proceedingsb@royalsociety.org

Reviewer(s)' Comments to Author:

Referee: 1

Comments to the Author(s)

This manuscript reviews (mainly) the theoretical aspects of niche width evolution in host parasite systems. It focusses on viral infections, which in current times is highly relevant due to its direct implications in zoonosis. It's main strenght is that it nicely illustrates the differences between eco-evolutionary and population genetic models, and calls for uniting eco-evolutionary models with

population genetic theory. In general, I found reading this piece quite enjoyable and gained some new perspective on niche breadth evolution. Some terminology used and background knowledge expected is however rather addressed at the specialist and can probably leave the generalist a bit puzzled. I have made a few suggestions where this could be improved and have some other comments that can easily be included in a revision of the manuscript.

1.) Since I am not a theoretician, I cannot really judge the completeness of their model and theory overview. But I have the feeling that some ecological/demographic aspects of the genetic models for coevolution could be added. Gokhale, C.S., A. Papkou, A. Traulsen, and H. Schulenburg, Lotka-Volterra dynamics kills the Red Queen: population size fluctuations and associated stochasticity dramatically change host-parasite coevolution. *BMC Evolutionary Biology*, 2013. 13: 254. would make a valuable addition to the citation in l. 174.

2.) To me it seemed as if one assumption of most models is that they are somehow symmetric, i.e. there are 2 specialists (one for each host species) and 1 generalist. What happens when this symmetry is broken, i.e. the generalist can infect both host species, but there is only one specialist infecting one of the hosts? I could see that this would reduce the cost of generalism since many more infectable hosts are available.

3.) The conclusion seems to be rather short and reads more like a summary. Maybe adding a bit of outlook to what could be achieved practically from combining the different theories might give the whole article more weight?

Other suggestions:

-1195-201 While can guess the reasoning what is meant by 'resistance-infectivity range matching' and why this could be unstable for hosts, one or two explanatory sentences would really help to understand this important point.

-1199-201 it is a bit unclear what diversity is maintained here. I assume it is diversity of range breadth, but it could also be host and parasite diversity itself, which could also maintain range breadth. Can the nestedness of both levels of diversity mad a bit clearer?

-1209 I wonder if a benefit to specialization is not the other side of the same cost as the cost to generalism, since fitness is measured relative to each other. Can either exist without the other?

-box1: Could the 'where constraints between phenotypes emerge'-part of Pareto front be explained better? I have a hard time imagining what constraints between phenotypes are.

-1271 I somehow miss a section that details the rich body of empirical work on niche breadth. Box 2 only gives examples on trade-off and mutation accumulation, while the empirical data largely comes from other reviews. Maybe giving some more examples of how cost of generalism varies in detail for some systems might be helpful.

Referee: 2

Comments to the Author(s)
Please see the attached file.

Author's Response to Decision Letter for (RSPB-2020-1230.R0)

See Appendix B.

Decision letter (RSPB-2020-1230.R1)

22-Jul-2020

Dear Ms Visher

I am pleased to inform you that your manuscript entitled "The Problem of Mediocre Generalists: Population Genetics and Eco-Evolutionary Perspectives on Host Breadth Evolution in Pathogens" has been accepted for publication in Proceedings B.

If you are likely to be away from e-mail contact during this period, let us know. Due to rapid publication and an extremely tight schedule, if comments are not received, we may publish the paper as it stands.

Your article has been estimated as being 9 pages long. Our Production Office will be able to confirm the exact length at proof stage.

Open access

You are invited to opt for open access via our author pays publishing model. Payment of open access fees will enable your article to be made freely available via the Royal Society website as soon as it is ready for publication. For more information about open access publishing please visit our website at http://royalsocietypublishing.org/site/authors/open_access.xhtml.

The open access fee is £1,700 per article (plus VAT for authors within the EU). If you wish to opt for open access then please let us know as soon as possible.

Paper charges

Sincerely,
Proceedings B
<mailto:proceedingsb@royalsociety.org>

Appendix A

Generalists and specialists have been of great interest in ecology and evolution for decades. This is exemplified by the fact that different subfields have attempted to tackle this topic. However, none have attempted to bridge the gap between models in population genetics and disease ecology. The authors say: “These sub-fields’ broad perspectives towards infectious disease evolution differ in several key ways and are not often well integrated, especially across intra- and inter-host scales (Geoghegan and Holmes, 2018; Mideo et al., 2008). Both population genetics and eco-evolutionary theory have important insights for why generalists are not ubiquitous, but these perspectives have not been well combined for a unified understanding of viral niche breadth evolution (Kawecki, 1994; Levins, 1968; Osnas and Dobson, 2012; Regoes et al., 2000; Remold, 2012; Whitlock, 1996), though see (Ogbunugafor et al., 2010).” This sums up their thesis, and I strongly agree. I believe that others in the field will as well.

What I appreciated about this paper is that it got into the different approaches of these models and how they relate to each other. It integrates concepts from both fields, making it especially useful to people who are new to the field or who specialize in one field and have passing knowledge in the other. It also does a good job of covering topics that are often missed in other papers on this topic, e.g. studies of large population sizes often ignore the role of mutation accumulation and pareto fronts in general. Overall, the prose was very readable, and clearly, the authors have carefully crafted the text to explain the underlying issues and models. I personally found this paper edifying and an interesting read.

Here are a few points that I’d like to see addressed to increase clarity:

I noticed that the authors sometimes refer to pathogens, sometimes virus, sometimes parasites. Does it matter or do the authors use the terms interchangeably? Can they clarify why they use one term over another in different instances? Does this term matter for the models?

The authors mention Pareto fronts but only in asides or boxes, e.g. line 94. I would love to see more explanation of what exactly they mean and when the concept has been used to explain costs of generalism.

Also, the paragraph starting on line 165 is a bit harder to understand. Would the authors please spell out when they mean or more generally elaborate on the concepts described here?

Here are a few minor line-by-line comments:

Line 75: Why “therefore”? It doesn’t seem to obviously follow from the previous point. I’m in favor of just omitting the word.

Figure 1: Goal? Goal of what? The models at those scales? The goal of viruses at those scales? “Hosts may heterogenous.” I agree. Do the authors mean variation within a species or different species entirely? Or, does this even really matter?

Line 159: add the “(GfG)” abbreviation here so it can be called back later in the paragraph.

Line 160: I believe that “trade-off” should not be hyphenated here. My understanding is that the hyphen is appropriate for the noun but not the verb form, as is used here.

<https://www.merriam-webster.com/dictionary/trade-off> Not trying to nitpick here; I had to reread the sentence a couple of times until I realized what was being said.

Line 165: the authors no longer use the Gfg notation. Would they please standardize this throughout the text?

Line 189-91: something is off with the wording of this sentence. Maybe add a comma after “pathogen resistance?”

Line 248-9: “...but also evolve as an indirect consequence of specialization due to mutational accumulation.” What do the authors mean by this? The conditions that specialists evolve in naturally incline them toward mutation accumulation? Also, note that sometimes the authors refer to mutation or mutational accumulation.

Box 3: When the authors refer to tight bottlenecks in cell cultures, are they referring to the population size when transferring a virus to a new culture? Or, does the tissue culture comment not translate to the tight bottleneck?

Appendix B

Response to Reviewers

Thank you all so much for your kind comments on this manuscript. We are pleased that you found the material engaging and are grateful for your suggestions of several important avenues for improvement.

Detailed below are responses to all of the comments.

Reviewer 1:

This manuscript reviews (mainly) the theoretical aspects of niche width evolution in host parasite systems. It focusses on viral infections, which in current times is highly relevant due to its direct implications in zoonosis. It's main strength is that it nicely illustrates the differences between eco-evolutionary and population genetic models, and calls for uniting eco-evolutionary models with population genetic theory. In general, I found reading this piece quite enjoyable and gained some new perspective on niche breadth evolution. Some terminology used and background knowledge expected is however rather addressed at the specialist and can probably leave the generalist a bit puzzled. I have made a few suggestions where this could be improved and have some other comments that can easily be included in a revision of the manuscript.

Thank you for these comments. We hope that the below changes will help to make the manuscript more accessible to generalists.

1.) Since I am not a theoretician, I cannot really judge the completeness of their model and theory overview. But I have the feeling that some ecological/demographic aspects of the genetic models for coevolution could be added. Gokhale, C.S., A. Papkou, A. Traulsen, and H. Schulenburg, Lotka–Volterra dynamics kills the Red Queen: population size fluctuations and associated stochasticity dramatically change host-parasite coevolution. BMC Evolutionary Biology, 2013. 13: 254. would make a valuable addition to the citation in l. 174.

Thank you for suggesting this citation. We originally left this paper out as it discusses matching allele dynamics – where all hosts and parasites are specialists - and thus does not include variation in range breadth. We included Song et al. 2015 with the same co-authors, which also allows for variation in breadth. Since Song et al. 2015 clearly builds on Gokhale et al. 2015 though, we agree that we should include both. We have also added an extra sentence about how demography impacts these models. The section now reads “However, population genetics models typically assume constant large population sizes and extending these gene-for-gene models to include ecology shows that demographic and ecological factors can strongly affect dynamics of a system (Ashby and Boots, 2017; Frank, 1991; Gokhale et al., 2013; Song et al., 2015).

Allowing fluctuations in population size disrupts the regularity of cycling and can lead to dampening over longer time scales.”

2.) To me it seemed as if one assumption of most models is that they are somehow symmetric, i.e. there are 2 specialists (one for each host species) and 1 generalist. What happens when this symmetry is broken, i.e. the generalist can infect both host species, but there is only one specialist infecting one of the hosts? I could see that this would reduce the cost of generalism since many more infectable hosts are available.

This is a good point and would depend on the assumptions of the model. Nested gene-for-gene or range models often have one specialist and one generalist where the second host would only be resistant to infection if it has a wider resistance breadth. In this case, the resistance breadth is costly, so the host reproduces slower and will evolve a new strategy when the parasite evolves to infect it (see discussion of Best 2010). However, if the second host is resistant to the specialist simply because it is different than the first, a generalist would have more infectable hosts. It is true that models of this type do not often account for ‘empty niches’, though empirical experiments sometimes do and show that generalists dominate these systems (see Lisa Bono’s 2013 and 2015 papers). However, under longer evolutionary time scales, we would expect this empty niche to be filled with a specialist of its own. Therefore, the lowered cost of generalism would be temporary until the niche is filled, although certainly with the potential to be important in the shorter term. Moreover, many natural systems may be unstable with periodic extirpations of parasites that lead to empty hosts. In this case, the system might not regularly have time to evolve to fill empty niches with specialists and generalists may be favored both due to the increased likelihood that they don’t go extinct from host loss and due to their increased access to empty host niches. There are some analogues here to conversations about the increased extinction probability of specialists in the macroevolutionary and conservation biology literatures (Clavel et al. 2011 and Colles et al. 2009).

We now include this discussion in several places.

Line 225- “These models generally assume symmetry so that there is either one specialist per host or a single generalist population, but empirical results have also shown that the presence of a host without a specialist (ie. an empty niche) can select for generalists (Bono et al., 2015). However, this benefit is likely to be temporary as empty niches are likely to be filled by their own specialists (MacArthur and Levins, 1967).”

Line 297- “However, macroevolutionary trends suggest that specialist lineages have higher extinction rates than generalists and prefer to specialize on more stable environments (Colles et al., 2009; Kammer et al., 1997; Sasal et al., 1999).”

3.) The conclusion seems to be rather short and reads more like a summary. Maybe

adding a bit of outlook to what could be achieved practically from combining the different theories might give the whole article more weight?

Thank you for this suggestion, we have added several sentences and a new paragraph to the end of the conclusion. It now reads, “Unifying these perspectives to explore how non-trade-off costs of generalism shape the co-evolution of hosts and pathogens in eco-evolutionary models would allow us to determine whether the mechanism of costly generalism shapes niche breadth co-evolutionary dynamics in ways that alter our understandings about the ecological conditions predicted to drive spillover and maintain diversity. It is likely that differences, if they arise, are likely to depend on temporal scale and open up further avenues of research regarding the role of temporal scale and habitat instability in niche breadth diversification dynamics (Ferris et al., 2020; Lenski and May, 1994).

Finally, the lack of unification across population genetics and eco-evolutionary perspectives is not limited to niche breadth evolution. A wider consideration of the exact nature of trade-offs could help understand how selective pressures are maintained across scales. For example, the addition of population genetics processes into evolutionary epidemiology models on the virulence and transmission trade-off provides predictive insights beyond those generated by simple trade-off functions (Day and Gandon, 2007; Day and Proulx, 2003). This may be particularly important since the pathogen particles that transmit are not necessarily the same ones causing host damage (Ebert and Bull, 2003), meaning that both the mean and the variance of the genotypes within the host determine the outcome of pathogenesis and transmission for the infection. In general, a better consideration of the nature of trade-offs in light of constant mutation pressure and ecological feedbacks should help us better understand ecological and evolutionary processes.”

Other suggestions:

-l195-201 While can guess the reasoning what is meant by 'resistance-infectivity range matching' and why this could be unstable for hosts, one or two explanatory sentences would really help to understand this important point.

We have added an extra sentence to clarify. It reads, “In other words, a host whose resistance breadth exactly overlaps with a parasite’s infectivity breadth should either evolve further costly resistance to prevent infection from that parasite or evolve less resistance to reproduce faster since it is susceptible anyway.”

-l199-201 it is a bit unclear what diversity is maintained here. I assume it is diversity of range breadth, but it could also be host and parasite diversity itself, which could also maintain range breadth. Can the nestedness of both levels of diversity mad a bit clearer?

It is both. The models start with a monomorphic population that branches as range breadth evolves. Because there are then more host and parasite strains, the system continues to evolve new range breadths and branch. Thus, there is a diversity of host and parasite strains with different range breadths. In these models, range breadths are entirely nested though, so matching allele dynamics (as talked about in the population genetics section) do not occur. We have clarified this in the sentence introducing these models, “Best et al. (2010) shows that a co-evolutionary range model, where resistance and infectivity breadths are nested and costly, will generate and maintain stable diversity with co-existing hosts and parasites across the generalism-specialism range.”

-I209 I wonder if a benefit to specialization is not the other side of the same cost as the cost to generalism, since fitness is measured relative to each other. Can either exist without the other?

We were not clear that here we were referring to a benefit compared to the ancestral state. So, a benefit to specialization means that a specialist can more effectively evolve increased fitness on one environment without necessarily decreasing its fitness on alternate environments. We have edited this sentence to read, “Effectively, if there is a benefit to specialization compared to the ancestor there does not need to be a cost to generalism.”

-box1: Could the 'where constraints between phenotypes emerge'-part of Pareto front be explained better? I have a hard time imagining what constraints between phenotypes are.

We have rephrased the definition in Box 1. It now reads, “The optimal trade-off front dividing accessible, suboptimal phenotype space from phenotype space that is inaccessible due to constraints”. We have also provided more details about Pareto fronts in the body of the text. See response to reviewer 2.

-I271 I somehow miss a section that details the rich body of empirical work on niche breadth. Box 2 only gives examples on trade-off and mutation accumulation, while the empirical data largely comes from other reviews. Maybe giving some more examples of how cost of generalism varies in detail for some systems might be helpful.

We have added a number of experimental studies looking at parasite evolution in homogenous and heterogenous host populations since these are not well discussed in the review papers on trade-offs. We have added “Empirical results do suggest that experimental evolution along genetically diverse hosts can constrain the evolution of virulence or the rate of adaptation (Cornwall et al., 2018; Ebert, 1998; González et al., 2019; Hughes and Boomsma, 2004; Kubinak

et al., 2015).”

Reviewer 2

Generalists and specialists have been of great interest in ecology and evolution for decades. This is exemplified by the fact that different subfields have attempted to tackle this topic. However, none have attempted to bridge the gap between models in population genetics and disease ecology. The authors say: “These sub-fields’ broad perspectives towards infectious disease evolution differ in several key ways and are not often well integrated, especially across intra- and inter-host scales (Geoghegan and Holmes, 2018; Mideo et al., 2008). Both population genetics and eco-evolutionary theory have important insights for why generalists are not ubiquitous, but these perspectives have not been well combined for a unified understanding of viral niche breadth evolution (Kawecki, 1994; Levins, 1968; Osnas and Dobson, 2012; Regoes et al., 2000; Remold, 2012; Whitlock, 1996), though see (Ogbunugafor et al., 2010).” This sums up their thesis, and I strongly agree. I believe that others in the field will as well. What I appreciated about this paper is that it got into the different approaches of these models and how they relate to each other. It integrates concepts from both fields, making it especially useful to people who are new to the field or who specialize in one field and have passing knowledge in the other. It also does a good job of covering topics that are often missed in other papers on this topic, e.g. studies of large population sizes often ignore the role of mutation accumulation and pareto fronts in general. Overall, the prose was very readable, and clearly, the authors have carefully crafted the text to explain the underlying issues and models. I personally found this paper edifying and an interesting read.

Thank you so much for your comments.

Here are a few points that I’d like to see addressed to increase clarity:

I noticed that the authors sometimes refer to pathogens, sometimes virus, sometimes parasites. Does it matter or do the authors use the terms interchangeably? Can they clarify why they use one term over another in different instances? Does this term matter for the models?

We have used pathogen and parasite relatively interchangeably and used virus where the paper specifically deals with viral evolution or the concept depends on nuances of viral biology. In many of the models, the parasite and pathogen are treated interchangeably. We have added an entry to the glossary for “Parasite and Pathogen” that reads “Parasite and pathogen both refer to organisms that are dependent on hosts for replication and cause host damage; we have used these terms interchangeably. We use virus when talking about empirical virus data or concepts that depend on nuances of viral biology.”

The authors mention Pareto fronts but only in asides or boxes, e.g. line 94. I would love to see more explanation of what exactly they mean and when the concept has been used to explain costs of generalism.

*We have added several sentences to the discussion of Pareto fronts, including more detail about how the concept has been used. The section now reads, “Li et al. (2019) showed that metabolic functions in yeast trade off with specialists having the highest performance on each metabolic function and an optimal trade-off front (ie. **Pareto front**) defining a region of inaccessible no-cost generalist parameter space (Shoval et al., 2012). They also showed that many genotypes can exist below the Pareto front. In this case, no-cost generalist alleles could advance the fitness of these maladapted genotypes to **Pareto front**, but evolution would be determined by the trade-off afterwards (Li et al., 2019; Shoval et al., 2012).”*

Also, the paragraph starting on line 165 is a bit harder to understand. Would the authors please spell out when they mean or more generally elaborate on the concepts described here?

*We have added several sentences to clarify the concepts in this paragraph. It now reads:
“However, classic gene-for-gene models only allow fluctuations in range breadth, (i.e. the number of hosts that a parasite infects or the number of parasites that the host resists) (Ashby and Boots, 2017). This is because they imply that the resistance and infectivity ranges of their hosts and parasites are entirely nested so that the least infective parasite only infects the most permissive host and the most resistant host can only be infected by the most infective, generalist parasite. Instead, gene-for-gene interactions may coexist with matching allele interactions with multiple specialists on a spectrum or as part of a two part process to allow for rare genotype advantage (Agrawal and Lively, 2002, 2003). When the assumption of complete nestedness is relaxed so that there can be multiple identically ranged hosts and parasites infecting different subsets of the total population, both high frequency cycling between hosts and parasites matching different subsets of the range and lower frequency cycling in range breadth can occur (Ashby and Boots, 2017). In other words, identities of the hosts and their matching parasites in the system will cycle quickly due to rare genotype advantage and the average range breadth of the hosts and their parasites will cycle more slowly due to costs of range breath. However, population genetics models typically assume constant large population sizes and extending these gene-for-gene models to include ecology shows that demographic and ecological factors can strongly affect dynamics of a system (Ashby and Boots, 2017; Frank, 1991; Gokhale et al., 2013; Song et al., 2015). Allowing fluctuations in population size disrupts the regularity of cycling and can lead to dampening over longer time scales.”*

Here are a few minor line-by-line comments:

Line 75: Why “therefore”? It doesn’t seem to obviously follow from the previous point. I’m in favor of just omitting the word.

*We have deleted the word “therefore”. The sentence now reads, “With **temporal** environmental heterogeneity, costs to generalism can select for single (**fine grained**) or multiple (**coarse grained**) specialists (Levins, 1968).”*

Figure 1: Goal? Goal of what? The models at those scales? The goal of viruses at those scales? “Hosts may heterogenous.” I agree. Do the authors mean variation within a species or different species entirely? Or, does this even really matter?

We meant to imply the goals of viruses at each scale. We have changed the language from “Goal” to “Fitness considerations” to clarify. We meant to convey that there are many axes and amounts of heterogeneity between hosts. For these purposes, the differences between intra- and interspecific variation doesn’t really matter. We have changed this sentence to “Hosts have many axes of heterogeneity”.

Line 159: add the “(GfG)” abbreviation here so it can be called back later in the paragraph.

We have added the GfG abbreviation. The phrase now reads, “the main model for host range in host-pathogen systems is the gene-for-gene (GfG) model”

Line 160: I believe that “trade-off” should not be hyphenated here. My understanding is that the hyphen is appropriate for the noun but not the verb form, as is used here. <https://www.merriam-webster.com/dictionary/trade-off> Not trying to nitpick here; I had to reread the sentence a couple of times until I realized what was being said.

We have corrected “trade-off” to “trade off”. The sentence now reads, “infectivity and resistance ranges directly trade off with pathogen replication and host reproduction respectively”

Line 165: the authors no longer use the Gfg notation. Would they please standardize this throughout the text?

We have edited the text to consistently use the “gene-for-gene” notation. We have kept the GfG notation only in the sentence where we introduce the gene-for-gene model with “the main model for host range in host-pathogen systems is the gene-for-gene (GfG) model”.

Line 189-91: something is off with the wording of this sentence. Maybe add a comma after “pathogen resistance?”

Thank you for catching this. We have added the comma and another “that” to the second clause. The sentence now reads “It is also clear that hosts commonly evolve pathogen resistance, and that different ecological dynamics between host and pathogen populations are likely to create varying selection pressures for resistance evolution and, therefore, co-evolutionary feedbacks”.

Line 248-9: “...but also evolve as an indirect consequence of specialization due to mutational accumulation.” What do the authors mean by this? The conditions that specialists evolve in naturally incline them toward mutation accumulation? Also, note that sometimes the authors refer to mutation or mutational accumulation.

We have changed all references to mutation accumulation. We have also edited this line and added an extra sentence to clarify our statement. This section now reads, “In a 2005 review, Maclean briefly considers trade-off and mutation accumulation models and suggests that there may not actually be a conflict between these theories if we consider that trade-offs may occur not only as a result of negative genetic correlations, but also evolve as an indirect consequence of mutation accumulation during specialization. In other words, the accumulation of environmentally specific deleterious alleles could result in fitness correlations that function analogously to genetic trade-offs in eco-evolutionary contexts.”

Box 3: When the authors refer to tight bottlenecks in cell cultures, are they referring to the population size when transferring a virus to a new culture? Or, does the tissue culture comment not translate to the tight bottleneck?

*We were referring to transmission bottlenecks in natural populations, not cell culture. We have rephrased this sentence for clarification. It now reads, “However, recent work in natural host-pathogen systems has shown increasing consideration about how these within-host evolutionary processes translate to selection between hosts through the extremely tight, **drift** promoting bottlenecks of many natural transmission events”*